# An Artificial Lens Capsule with a Lens Radial Stretching System Mimicking Dynamic Eye Focusing

**DOI:** 10.3390/polym13203552

**Published:** 2021-10-15

**Authors:** Huidong Wei, James S. Wolffsohn, Otavio Gomes de Oliveira, Leon N. Davies

**Affiliations:** 1College of Health and Life Sciences, Aston University, Birmingham B4 7ET, UK; l.n.davies@aston.ac.uk; 2Rayner Intraocular Lenses Limited, Worthing BN14 8AQ, UK; otaviogomes@rayner.com

**Keywords:** presbyopia, accommodation, AIOL, ALC, LRSS

## Abstract

Presbyopia is a common eye disorder among aged people which is attributed to the loss of accommodation of the crystalline lens due to the increasing stiffness. One of the potential techniques to correct presbyopia involves removing the lens substance inside the capsule and replacing it with an artificial lens. The development of such devices, e.g., accommodating intraocular lenses (AIOLs), relies on the understanding of the biomechanical behaviour of the lens capsule and the essential design verification ex vivo. To mimic the eye’s dynamic focusing ability (accommodation), an artificial lens capsule (ALC), from silicone rubber accompanied by a lens radial stretching system (LRSS) was developed. The ALC was manufactured to offer a dimension and deforming behaviour replicating the human lens capsule. The LRSS was calibrated to provide a radial stretch simulating the change of diameter of capsules during accommodating process. The biomechanical function of the ALC was addressed by studying its evolution behaviour and reaction force under multiaxial stretch from the LRSS. The study highlighted the convenience of this application by performing preliminary tests on prototypes of ophthalmic devices (e.g., AIOLs) to restore accommodation.

## 1. Introduction

Presbyopia is the loss of the dynamic focus and accommodating amplitude usually being more significant for people after their middle 40s. It is largely attributed to the ineffectiveness of accommodation due to changes in the natural crystalline lens [1,2,3,4,5]. The Helmholtz theory is a widely accepted explanation of the accommodation of human eyes [6,7]. Under the accommodated state for near vision (Figure 1A), the ciliary muscle contracts, moves forward, releasing the zonular tension allowing the lens returns to more curved form, achieving high optical power. When the ciliary muscle releases the contraction (Figure 1B), the zonules become tensioned and the lens is stretched with flattened surfaces, reducing the optical power for distance vision, and arriving at the unaccommodated state. The cause of presbyopia primarily results from the increased stiffness of the lens due to age, which has been demonstrated through experimental and numerical modelling [4,8,9]. Studies of the biomechanical behaviour of the lens capsule, however, showed no evident degradability of capsule with age [10,11,12,13]. The accommodating function of the ciliary muscle is largely maintained even after the removal of natural crystalline lens [14,15]. These studies suggest a potential to restore accommodation by replacing the lens content with implantable devices, e.g., accommodating intraocular lenses (IOLs) or refilling of the capsular bag [16,17,18].

The lens capsule is a typical basement membrane with a thickness of 5 to 30 µm [10,13,19], which can be well maintained by using modern phacoemulsification techniques during eye surgery [20]. It moulds the lens substance when driven by the ciliary body as a result of its superior mechanical stiffness [11,19]. There is a hyper-elastic stress–strain relationship of the capsule with mechanical testing, demonstrating variability of thickness and stiffness depending on age and spatial distribution [10,11,12,19,21,22]. To understand the lens accommodating process, ex vivo testing has been widely employed to replicate the contraction of ciliary muscle by applying and releasing stretch in a mechanical way on ex vivo lenses with zonules and ciliary body attached [2,12,23,24,25,26], but lacking the independent investigation on the capsule itself. The difference in the biomechanics of the lens with and without substance exists by the different force response of the filled and empty capsular bag [12], where the latter demonstrates the status after surgery.

For early prototypes of accommodating IOLs or other mechanical techniques used to treat presbyopia, it is very challenging to perform the tests inside human eyes due to the limitations of resources, time commitment and ethical considerations [27]. Animal tissues have been used, but intrinsic differences exist between the dimensions and mechanical behaviour compared to a human lens [12,25,28]. In this study, an artificial lens capsule (ALC) from silicone rubber was suggested as an alternative to the biological lens capsule with a similar hyper-elastic behaviour [29]. The advantage of using ALC was highlighted by its manipulated dimension and deforming behaviour to mimic the human capsule. The base material selected was a synthesised material for easy accessibility, a wide range of stiffness and popular applications. Together with it, a customised lens radial stretching system (LRSS) was developed to test the deforming behaviour of ALC at conditions replicating accommodation. The results of profile change and reaction forces of ALC under accommodation provided a comparison to the human capsule and identified its further improvement and applications.

## 2. Materials and Methods

### 2.1. Preparation of Artificial Lens Capsule (ALC)

A room temperature vulcanised (RTV) industrial clear silicone (Grade: za13, Zhermack, Rovigo, Italy) was selected for the preliminary study with a focus on deforming behaviour. The cured sample had a Shore A Hardness of 13 with an equivalent Young’s modulus (of 0.5 MPa) by calculation [30,31], close to the low limit of stiffness of a human capsule [13,19]. The silicone material had a base fluid A and catalyst fluid B which was mixed thoroughly at a ratio of 1:1. The mixed fluids were vacuumed for three minutes to eliminate any air pockets. In the moulding process (Figure 2A), the degassed fluids were poured into a female mould with a concave shaped cavity. A male mould with convex shape moved downwards along the axis to form the shape of half of the capsule (anterior or posterior) with a uniform thickness of 150 µm. The extra silicone fluids were expressed out of the mould and formed a flat edge along the equator region with a thickness of 500 µm. At least 24 h were allowed for the curing of the material at room temperature (of 23 °C).

The cured anterior and posterior half was treated to have isotropic mechanical properties. They were assembled to form an artificial capsule (Figure 2B), which had the ellipsoidal shapes of capsules with a radius of 4.50 mm at the equator (E), sagittal distance of 2.55 mm at the anterior pole (A) and 1.60 mm at the posterior pole (P), respectively. The flat edge at the equatorial region had 1.50 mm width around the capsule body, i.e., a total diameter of 12.00 mm. A three-dimensional plastic support structure was manufactured by 3D printing to connect the ALC to the lens stretching system (Figure 2C), which had a circular region with internal diameter of 10.00 mm and external diameter of 12.00 mm to match the size of ALC. Eight extended branches of the support were designed to be connected to a lens stretching system and fixed around the holes with screws. Adhesive (Ethyl2-cranoa-crylate, Loctite control, Henkel, Winsford, UK) was applied evenly on the bottom side of the circular region of the support structure to form a thin layer, where the ALC was attached and centered before the curation of adhesive.

### 2.2. Stretching Test Mimicking Accommodation

An experimental setup mimicking accommodation was designed (Figure 3A), where the artificial lens capsule (ALC) was installed on a customised lens radial stretching system (LRSS) inside a water bath at room temperature. Two digital microscope cameras on the top (Model: Dinolite am73915mztl, AnMo, Taibei, Taiwan) and side (Model: cmex18pro, Euromex, Arnhem, The Netherlands) along with sufficient illumination were employed to monitor the process from two orthogonal directions. The imaging process was performed by using the respective software for each camera (Dinolite: DinoCapture 2.0, CMEX: ImageFocusAlpha).

A lens radial stretching system (LRSS) was developed to provide a radial stretch on the ALC by connecting the support structure with 8 arms. A pin-slot mechanism was employed to convert the rotation of gears to linear motion of the arms (Figure 3B). Before stretch, the lens capsule was defined at accommodated state with more curved form. During the stretching process, each arm moved simultaneously along the radial direction and applied stretch to the lens capsule to achieve the flatter surfaces of ALC, i.e., arriving at an unaccommodated state. Compared to previous similar applications [24], the bespoke LRSS offered a more precise motion by using a stepper motor (1.8° step angle, Model: 1705hs200a-nema 17, Ooznest, Brentwood, UK) together with a microcontroller and encoder (Model: S-lite, uStepper, Aalborg, Denmark). The motor was driven by a sub-step of 0.1° (3200 steps per revolution), resulting in a designed incremental linear step of 7 µm by the transmission. A submersible miniature S-Beam load cell (Model: Lsb 210, Futek, Irvine, CA, USA) was installed on one arm to measure the reaction force during stretch with a precision of 0.01 g and upper limit of 100 g (Figure 3C). The load cell was pre-calibrated by the supplier and in-situ force values were monitored and recorded by the accompanied software (Sensit, Futek, Irvine, CA, USA). To provide independent motion of each segment of the ALC, a radial cut was performed at the adjacent branches of support structure after mounting the ALC onto LRSS. As allocated in the support structure, a surgical knife with fine blade was used to perform the cutting from the edge of the structure until it reached the equatorial radius of the ALC.

One risk recognised during the mounting operation was the damage to the initial shape of the moulded ALC, causing a wrinkled capsule surface due to the collapse of the membrane (Figure 4A). This was similar to the situation occurring in clinical practice during the cataract surgery, where viscoelastics (OVDs) were usually employed to protect the capsule during the operation. This approach was also used to recover the shape of a collapsed ALC by pressurising it with OVDs (Ophteis, 1% Solidum Hyaluronate Concentration, Rayner, Worthing, UK), which recovered the smooth surface of the ALC after filling (Figure 4B). The weakness of this operation was a lack of pressure control during the filling process, possibly leading to over-inflation of the ALC compared to the initial moulded shape. To compare the influence from filling OVDs, the inflated ALC was imaged from the side view (camera 1) and measured. A fitted arc (red) of the designed ALC curve (empty) was compared to the fitted profile of the inflated ALC (OVDs filled) (Figure 4C). The fitting method was proved to be applicable, matching the designed profile of the ALC very well within the equatorial diameter of 6 mm. Comparing the fitted arcs before and after inflation, there was a more curved surface of the pressurised ALC on both anterior and posterior surface, implying an over-inflation condition. Therefore, the profile of the ALC after inflation was used as the reference by taking the images before the onset of radial stretch.

The LRSS with ALC mounted was calibrated preliminarily by using the camera (2) on the top to define a radial stretch of 0.5 mm at the equatorial edge by comparing the diameter of the inner rings of the support structure (Figure 4D). The sampling rate of the load cell was set at 5 ms by the Sensit software, i.e., 200 frames per second. Cyclic stretching and releasing the ALC can be observed, with the ALC held for 1 s after stretch (unaccommodated state) during the loading stage and 5 s residence was provided to allow the ALC to return to its initial shape when the stretch was released and ALC returned to zero displacement (accommodated state). Two linear nominal speed (NS) of 0.5 mm/s and 0.05 mm/s was defined for the stepper motor, which covered the range of the ex vivo tests of human lens [32,33,34]. The ALC was stretched primarily at a high speed (0.5 mm/s) then followed by the stretch at low speed (0.05 mm/s). The duration of effective elongation of ALC by calibration was found to be 2.7 s (NS = 0.5 mm/s) and 27 s (NS = 0.05 mm/s), respectively.

## 3. Results

### 3.1. Change of Diameter

At NS = 0.5 mm/s, the images were extracted from t = 0 to 2.7 s (Figure 5A). The diameter showed no evident change before t = 1.5 s whilst a diameter change was observed from t = 2.0 to 2.5 s. The quantitative data was used for calibration by averaging and plotting three-time traces (Figure 5C), showing three distinct stages during the stretching process. Before t = 1.0 s, there was a slow radial stretching process with speed of 0.01 mm/s. From t = 1.0 to 1.7 s, a transition stage followed with an increased speed of 0.12 mm/s. The total stretch from the preliminary two stages was less than 0.10 mm. The primary stretch occurred in the third stage, where there was a linear stretch at a speed of 0.30 mm/s and a final average displacement of 0.44 mm was reached, implying a radial deformation of 10% along the equatorial diameter. The deviations of three repeats on the first two stages appeared to be small and the ineffective stretch was attributed to overcoming the relaxed status of the ALC with support structure and tolerances within the LRSS assembly. High deviations were observed during the third stage due to the uncertainty of stretch and the dimensions of the ALC. There was a stable final stretching distance by a small deviation of the displacement at the end of stretch.

By comparing the shape evolution at low speed (NS = 0.05 mm/s) from at t = 0 to 27 s (Figure 5B), there was a similar tendency of no evident change of diameter before t = 15 s. This similarity showed the initial stretch required to exert sufficient tension on the ALC. An increase of diameter was observed from t = 20 to 27 s. Three different incremental stages were observed by processing the images at NS = 0.05 mm/s (Figure 5D), implying a linear stretch with speed of 0.001, 0.015 and 0.027 mm/s sequentially. A stretching distance of 0.075 mm was observed from the initial two stages. Compared to the stretch at high speed (NS = 0.5 mm/s), the deviations of displacement at low speed (NS = 0.05 mm/s) became smaller during the third stage due to more relaxation time, allowing the capsule to reach dimensional stability. The final average stretching distance reached 0.42 mm, similar to the result at high speed.

### 3.2. Change of Profile

The change of the profile was shown by imaging the deformation of ALC from the side view by the digital microscope camera 2. At high speed (NS = 0.5 mm/s) (Figure 6A), smooth capsule surfaces were observed with no obvious change before t = 1.5 s. More flattened anterior and posterior surfaces were found from t = 2.0 to 2.7 s. By measuring the surface curvature radius change with time, a typical shape evolution was plotted by fitting the profile with circular arcs (Figure 6C). The anterior surface showed similar curvature before t = 1.5 s whilst there was a minor increase of radius on the posterior surface. A significant increase of radius of curvature occurred on the anterior and posterior surface from t = 1.5 to 2.0 s whilst there was gradual shape change from t = 2.0 to 2.7 s.

By lowering the speed (NS = 0.05 mm/s) (Figure 6B), the evolution of the profile showed a similar behaviour: an indistinguishable change before t = 15.0 s and a flattened shape from t = 20 to 27 s. Compared to the smooth surfaces in Figure 6A, the initial profile (t = 0 s) showed more inhomogeneous anterior and posterior surfaces. This indicated the influence of cyclic loading from the previous stretch, introducing the partial loss of pressurisation of the ALC. As the radial stretch increased (t = 25 to 27 s), smooth surfaces were observed and there was no big difference with the previous loading condition. By fitting the curvature of anterior and posterior surfaces with circular arcs (Figure 6D), there was a similar gradual increase of curvature radius of the posterior surface from t = 15 to 27 s. In contrast to the steep increase of curvature radius at high speed, the profile of the anterior surface at low speed increased more continuously. Under the same equatorial radius of 4.5 mm, the final closed anterior and posterior shape suggested a low ultimate curvature radius. 

To compare the results at different speeds by applying stretch in sequence, the statistical deformation properties of ALC were summarised by three repeats at each condition (Table 1). Under an accommodated state before stretch (NS = 0.05 mm/s), the lens diameter increased on average 1.0 mm and higher deviation existed in the second loading condition. Due to the inhomogeneous curvature, the measured radius showed decreased average values, but increased deviations at the anterior and posterior surfaces, implying a more localised curved shape around the sagittal region. The sagittal distance showed decreased values due to the pressure loss on the capsule by the preliminary cyclic loading (NS = 0.5 mm/s).

After the ALC arrived at the unaccommodated state (Table 1), it can be observed that there were some differences on the final dimensions, but very similar results on the shape change. The dimensions at NS = 0.05 mm/s showed greater lens diameter, lower anterior and posterior surface curvature radius, and sagittal distance in contrast to the results at NS = 0.5 mm/s, which corresponded to the difference in initial profile. By comparing the shape change before and after stretch, there was a slight increase of lens diameter (of 0.035 mm) by stretching at lower speed (NS = 0.05 mm/s). In terms of the change of the curvature radius, an increase of 0.20 mm and decrease of 0.07 mm was found at the anterior and posterior surface, respectively. The sagittal distance at low speed increased by 0.07 mm compared to the result at high speed.

### 3.3. Reaction Force

Alongside the shape change, the reaction force reaction represents the mechanical response of the capsule under accommodating process. The repeatability of force measurement was shown by the cyclic readings of one typical measurement at NS = 0.5 mm/s (Figure 7A) and NS = 0.05 mm/s (Figure 7B). After the first cycle, the load cell reported a negative reading of −2.5 g, implying a compressive status at zero strain due to the stress softening of rubber materials, i.e., Mullins effect [35,36,37,38]. During the recovery stage lasting for 5.0 s before the next stretch, there was a gradual increase of the reaction force with slight difference at each cycle because of the relaxation behaviour of molecules. In a typical loading process after the first cycle, the onset of stretch was indicated by a decrease of force reading whilst the peak force reading indicated the arrest of stretch. There was a gradual decrease of the force reading at the recovery stage for 1 s due to relaxation. The release of stretch introduced a slight increase of force reading and it stopped when the force reading reached the lowest value (of −2.5 g).

The previous force readings implied the complex dynamic effect of the artificial capsule during the cyclic stretch and release. Corresponding to the increase of diameter at the loading stage, the reaction force of three tests at NS = 0.5 mm/s (Figure 7C) and 0.05 mm/s (Figure 7D) was compared by offsetting the initial reading to zero at the onset of stretch. Four stages were evident, with different tendency exhibited from the reaction force, defined as mobilisation (1), pretension (2), acceleration (3), and steady stretch (4). During the mobilisation stage, the load cell gave a force reading of 1.0 g, awaiting the stepper motor to arrive at the maximum velocity within t = 0.04 s (Figure 7C) and t = 0.4 s (Figure 7D). A pretension stage followed with a constant force reading, implying a blank stretch on the capsule, but only absorbing the relaxed state of capsule and the manufacturing tolerance, lasting for 1.0 s (NS = 0.5 mm/s) and 10.0 s (NS = 0.05 mm/s), respectively. The similar reaction force (of 1.0 g) for the previous two stages irrespective of the stretching speed indicated a process without the involvement of the ALC. In the acceleration stage, the force reading showed a sharp linear increase to 3.0 g approximately within t = 1.5 s (Figure 7C) and t = 15 s (Figure 7D). As the previous results showed a minor change of diameter (Figure 5) and profile (Figure 6) of the capsule, the force contribution was not primarily from the deformation, but from the damping effect due to the increase of velocity of the ALC. A steady stretching stage was presented after a constant velocity was reached, which was 0.3 mm/s and 0.027 mm/s, respectively. A slow force increase was observed at this stage related to the speed, displaying an average final maximum force of 7.12 ± 0.10 g (Figure 7C) and 5.76 ± 0.12 g (Figure 7D), respectively. Excluding the contribution of the damping force, the maximum force response from stretching of the capsule from one arm was approximately 3.78 ± 0.24 g (NS = 0.5 mm/s) and 2.83 ± 0.06 g (NS = 0.05 mm/s), respectively.

## 4. Discussion

The external dimensions of the ALC followed the measured biometry of the human capsule in vivo, with a diameter between 9.0 mm and 10.0 mm [12,23,39,40]; extraction of its contents can result in the increase of diameter of the capsule [12]. The application of OVDs to pressurise the ALC increased the sagittal distance slightly more than the biological capsule with values between 4.0 mm and 4.2 mm [12,23,39]. Compared to the external dimension, the ALC could only be fabricated with an even thickness (of 150 µm), whereas the thickness distribution of human capsules was very variable with dependence on the location and age [41,42,43]. The average thickness of anterior lens capsule was between 10 µm (0 year) and 20 µm (60 years), which had a Young’s modulus between 7.0 MPa (0 year) and 2.0 MPa (60 years) based on a bulge test [10,11]. In contrast, uniaxial tensile tests of the anterior capsule demonstrated a Young’s modulus between 0.4 MPa (1 year) and 2.5 MPa (60 years) with a thickness ranging from 15 µm (1 year) to 26 µm (60 years) [13]. The posterior capsule showed no strong dependence on age, with thickness ranging from 4 µm to 9 µm and having a similar mechanical behaviour to the anterior capsule [44]. Based on these studies, a cured silicone rubber with low Young’s modulus (of 0.5 MPa) was selected for the ALC membrane, aiming to provide an equivalent stiffness despite the greater thickness. The connection of the capsule to the human ciliary body was via zonules, located in a narrow but thick ring around the equatorial region close to the diameter of the ciliary ring (of 10 mm) [45]. It has been shown that the position of the zonules have no impact on accommodative dynamics as modelled with finite element analysis [46], but selective cutting of the zonules influences the contribution from the anterior and posterior capsule [47]. Hence, the extended ring along the equator facilitated the radial stretch, with an equal involvement of the anterior and posterior capsule.

During the ex vivo test of the biological lens (capsule) mimicking accommodation, the sclera together with ciliary body, zonules and lens was bonded to the end parts of different lens stretching system [2,12,23,24,25,26]. The surrounding tissues around the lens were cut radially to eliminate the circumferential contribution and separate the accommodation of lens [32,33]. This concept was employed in the ALC by cutting the extended ring into eight independent segments around the equator. Most lens stretching systems employ a radial stretch distance of 2 mm of the rigid parts to provide the necessary dimensional change under accommodation [12,23,25,26,47], whilst the maximum diameter change of a human lens (in a 48 year old) is 0.2 mm without an accompanied change of optical power [25]. An evident change of lens optical power (6 D to 8 D) can be achieved by applying a maximum radial stretch of 0.8 mm (in a 19 year old) and 1.0 mm (in a 34 year old) around the ring of ciliary body [2]. An in vivo human MRI investigation by applying an accommodative stimulus between 0.1 D and 8.0 D, has shown the diameter change of the ciliary body to be between 0.5 mm and 1.2 mm, corresponding to a diameter change of the crystalline lens of between 0.3 mm and 0.6 mm [48]. Smaller values of lens diameter change (0.28 mm to 0.32 mm) have been observed by the same technique under an accommodative stimulus of 5 D to 7 D [45]. The current study used a large radial stretch to achieve a bigger change of diameter (of 0.9 mm), trying to achieve the shape change of the ALC for the noticeable optical power adjustment. The two nominal linear speeds were very similar to the other reported displacement-controlled ex vivo tests of between 0.025 mm/s [32,33] and 0.5 mm/s [34]. Compared to other tests with sequentially increased elongation [25,34], a continuous radial stretch was applied to measure the force reaction more accurately by preventing the stress relaxation [19,49].

The shape change of the human lens during accommodation has been studied by quantitative measurement ex vivo [47,50,51] and in vivo [45,48,52], showing large deviations and strong dependence on age. Under the transition from the accommodated state to the unaccommodated state, the human lenses of young people (aged 19 to 29 years) show a curvature radius change of anterior surface R_A_ = 7.0 to 12.0 mm and of posterior surface R_P_ = 5.0 to 6.1 mm with 8 D power change in comparison to R_A_ = 9.0 to 9.8 mm and of R_P_ = 6.0 to 6.1 mm for lenses in older individuals (aged 55 to 70 years) with a 2 D power change [45,52,53]. The initial shape of the ALC (R_A_ = 7.0 mm, R_P_ = 5.0 mm) before stretch showed a similar profile to the young human lens in an accommodated state. After applying the maximum radial stretch, the posterior surface of ALC showed a similar curvature (R_P_ = 6.0 mm), but no marked curvature changes on the anterior surface (R_A_ = 9.0 mm) as with human lenses. This implied the ALC had a small optical power change (of 2 D) based on the accommodation stimulation of a human lens [52]. During human accommodation processes, significant variability of the change in ciliary ring diameter of between 0.5 mm and 1.0 mm and a change of lens thickness of between 0.1 mm and 0.6 mm has been shown [39,48]. The shape changes of the ALC fell into this region with a similar change of diameter (of 0.8 mm) and sagittal thickness (of 0.6 mm).

Most ex vivo studies on the human lens with the zonules radially cut have required an overall force level of 3 g to 5 g to achieve an unaccommodated state independent of age [25,32,33,39], whilst higher contraction forces (from 8 g to 13 g) are also reported [2,12,46]. The extraction of the lens substance from within the capsule only slightly reduces the force required [12,32]. Compared to the human lens (capsule), a similar level of reaction force (of 3 g to 4 g) excluding the damping effect was found on one arm, which indicated a higher overall reaction force across the eight tension arms (of 20 g to 30 g) with the ALC to achieve a similar deformation. This was primarily attributed to the larger thickness of ALC (of 150 µm) by a similar level of growth (of eight times) compared to the human capsule (of 5 to 30 µm) [10,11,19]. It concluded the appropriateness of the stiffness for the silicone material (of 0.5 MPa) selected for ALC. Compared to the previous studies [33,46], the ALC response driven by LRSS varied with the force applied over time. There was a notable pretension stage due to the relaxation of the sample and the tolerance of the lens stretching system, which had also been reported by a previous study [39]. 

## 5. Conclusions

An artificial lens capsule (ALC) fabricated from silicone rubber has been developed to offer an alternative to testing using a human biological capsule, whilst a complementary lens radial stretching system (LRSS) has been built to mimic the dynamics of accommodation. The experimental results suggest similarity of deformation between the ALC and human lenses, despite the discrepancy of the force reaction due to the high deviation of thickness. This study has created easy access to a realistic lens capsule, which was not influenced by physiological variability, and can be manufactured with controlled stiffness and dimensions. The applicability of the LRSS with reliable motion mechanism and connecting structure was demonstrated by providing robust stretch and reaction force over repeated test. The developed approach provides advantages for the preliminary tests on prototypes accommodating intraocular lens or other implantable devices used for restoring accommodation to overcome presbyopia.

## Figures and Tables

**Figure 1 polymers-13-03552-f001:**
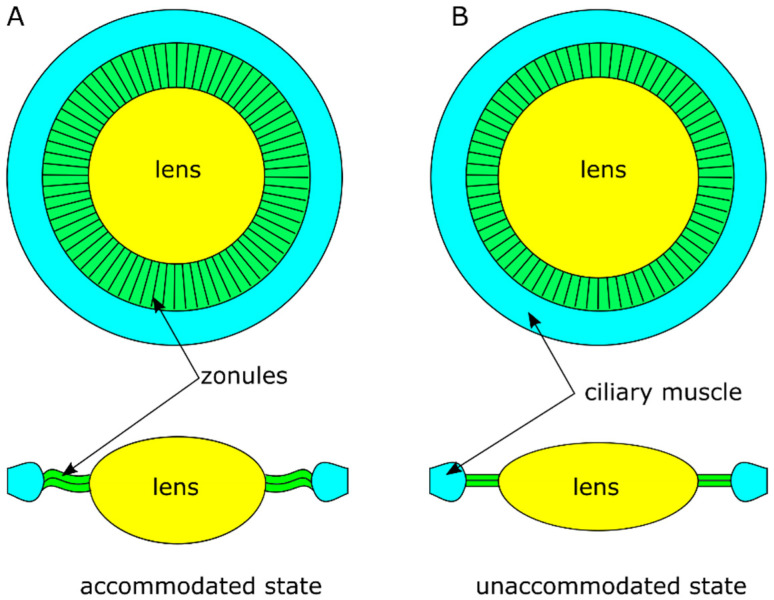
Natural accommodating process of eyes: (**A**) accommodated state for near vision; (**B**) unaccommodated state for distance vision.

**Figure 2 polymers-13-03552-f002:**
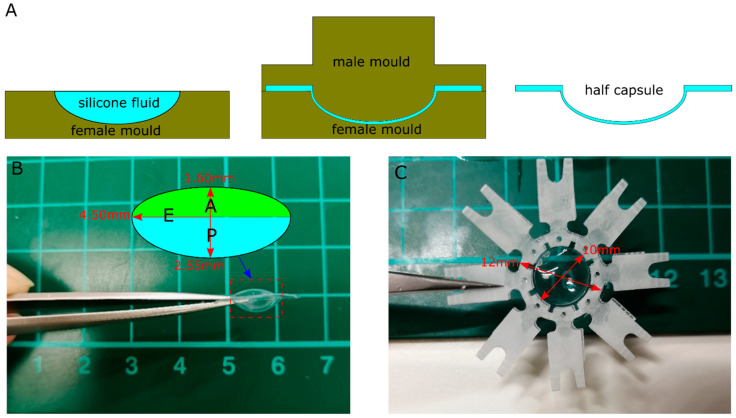
Preparation of artificial lens capsule (ALC): (**A**) moulding process; (**B**) assembled capsule; (**C**) with support structure.

**Figure 3 polymers-13-03552-f003:**
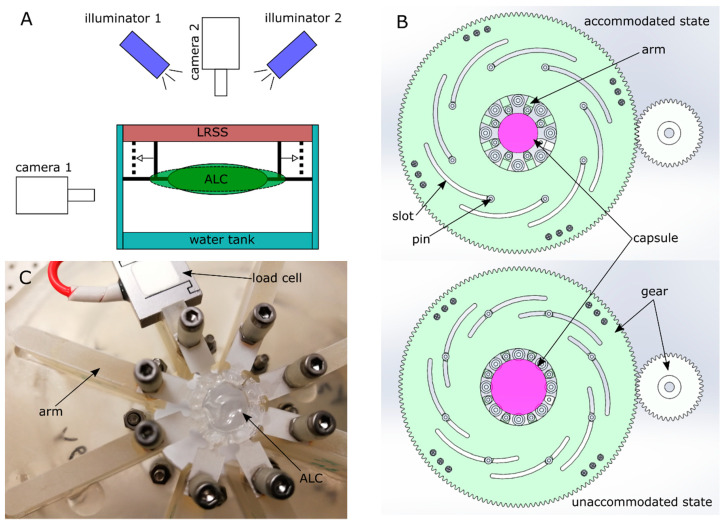
Experimental setup of the accommodating test: (**A**) illustration of the test scheme; (**B**) mechanism of the lens radial stretching system (LRSS); (**C**) integration of artificial lens capsule (ALC) to the LRSS.

**Figure 4 polymers-13-03552-f004:**
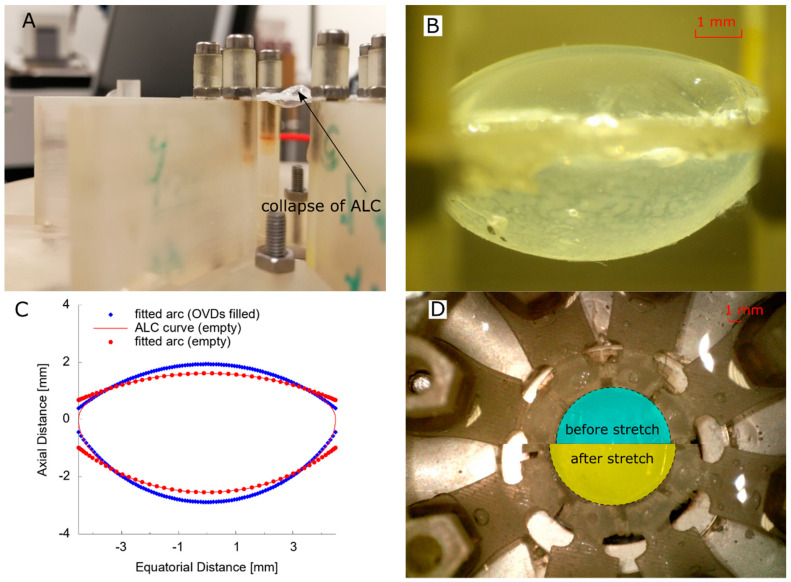
Calibration of LRSS with ALC: (**A**) ALC mounted on LRSS; (**B**) ALC on LRSS filled with OVDs; (**C**) comparison of ALC before and after filling OVDs; (**D**) comparison of lens diameter before and after stretch.

**Figure 5 polymers-13-03552-f005:**
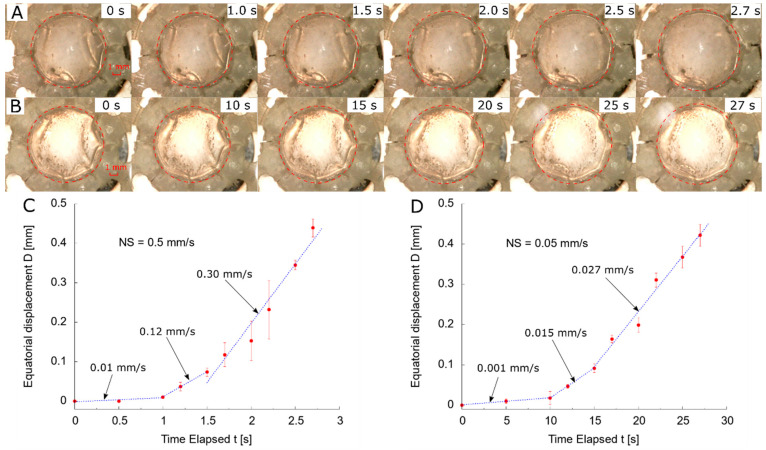
Stretching process from top view: (**A**) evolution of diameter at NS = 0.5 mm/s; (**B**) evolution of diameter at NS = 0.05 mm/s; (**C**) statistical displacement vs. time at NS = 0.5 mm/s; (**D**) statistical displacement vs. time at NS = 0.05 mm/s.

**Figure 6 polymers-13-03552-f006:**
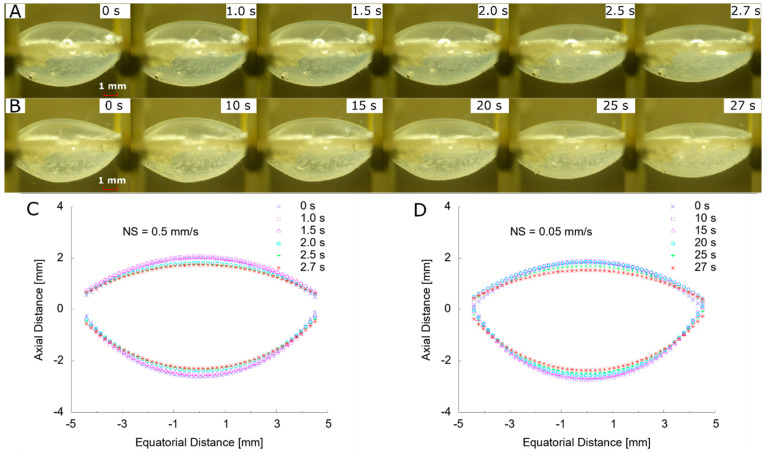
Stretching process from side view: (**A**) evolution of profile at NS = 0.5 mm/s; (**B**) evolution of profile at NS = 0.05 mm/s; (**C**) fitted profile at NS = 0.5 mm/s; (**D**) fitted profile at NS = 0.05 mm/s.

**Figure 7 polymers-13-03552-f007:**
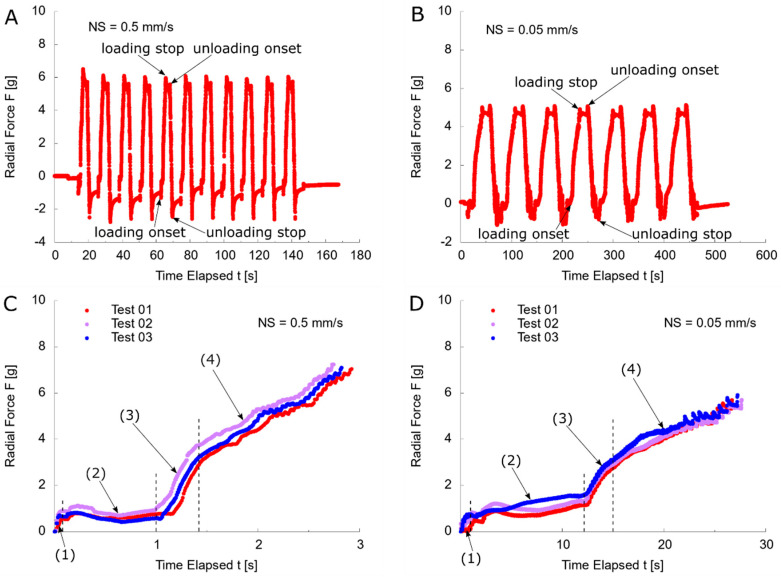
Force history of artificial capsule under radial stretch: (**A**) typical force reading during cyclic loading/unloading at NS = 0.5 mm/s; (**B**) typical force reading during cyclic loading/unloading at NS = 0.05 mm/s; (**C**) three repeated forces reading during loading process at NS = 0.5 mm/s; (**D**) three repeated forces reading during loading process at NS = 0.05 mm/s.

**Table 1 polymers-13-03552-t001:** Deforming properties of ALC from LRSS.

	Lens Diameter D [mm]	Curvature Radius		Sagittal Distance H [mm]
		$R_{A}\;[\mathrm{mm}]$	$R_{P}\;[\mathrm{mm}]$	
NS = 0.5 mm/s				
Accommodated	9.078 ± 0.003	7.338 ± 0.079	−5.360 ± 0.092	4.674 ± 0.031
Unaccommodated	9.921 ± 0.056	9.416 ± 0.079	−6.376 ± 0.053	4.063 ± 0.016
Increment	0.843 ± 0.055	2.079 ± 0.072	−1.015 ± 0.104	−0.610 ± 0.016
NS = 0.05 mm/s				
Accommodated	10.313 ± 0.054	7.010 ± 0.225	−4.853 ± 0.117	4.617 ± 0.014
Unaccommodated	11.073 ± 0.056	9.288 ± 0.186	−5.782 ± 0.126	3.937 ± 0.077
Increment	0.878 ± 0.045	2.278 ± 0.226	−0.929 ± 0.098	−0.680 ± 0.070

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
