# Peer review of "An Artificial Lens Capsule with a Lens Radial Stretching System Mimicking Dynamic Eye Focusing"

_polymers, 2021, doi:10.3390/polym13203552_

Round 1

Reviewer 1 Report

The manuscript investigated the biomechanical behavior of lens capsule and the essential design verification ex vivo. Hence, to mimic the dynamic eye focusing of eyes, an artificial lens capsule (ALC) from silicone rubber accompanied by a lens radial stretching system (LRSS) was developed. The subject of the manuscript is interesting and efficient in the field of biomechanics and could make an efficient contribution in this field. The paper is well-written, and the procedure is well expounded. Thus, the reviewer recommends publishing of the manuscript after addressing the following comments.

  1. As mentioned by authors, ALC is obtained by mixing two materials; so, does the final sample of ALC is isotropic homogeneous material? If yes, authors should mention it explicitly in the manuscript; if not, the Young’s moduli in different directions should be provided.
  2. How is the support structure connected to ALC? If it is connected via bolts, it may cause stress concentration on ALC which leads to inaccurate stress distribution on/ around ALC.
  3. Regarding the stretching process, section 3.1, it should be explicitly indicated that if all deformations under different speeds are in elastic region of the ALC material or it experienced elastic/ plastic deformation.
  4. Did authors intend to exhibit the fatigue test of ALC by representing Fig. 7? If yes, it should be explained why on the time axis of Fig. 7.A each cycle lasts about 20 seconds while on the time axis of Fig. 7.c only 3 seconds is displayed. This issue should be clarified and explained in the manuscript.
  5. The reviewer suggests checking the stretch speed in which the rupture of ALC will occur. The rupture test and the ultimate tensile strength of ALC is also of crucial importance.

Reviewer 2 Report

The current manuscript provides an interesting account of an artificial lens capsule with a lens radial stretching system. The study is novel and innovative and provides a unique strategy for mimicking dynamic eye focusing. The experiments are well conducted and the results are well reported. I recommend acceptance of the manuscript after following minor revisions:

1. The contact angle data and the vapour permeation data will be good addition to the work done as it will directly add value to the practicality of the lens.

2. I suggest authors to add morphological images/data for the surface of the lens.
